# Obtaining Land Cover Type for Urban Storm Flood Model in UAV Images Using MRF and MKFCM Clustering Techniques

**Yanmei Wang**, **Chengcai Zhang \***, **Yisheng Zhang**, **He Huang** and **Lingtong Feng**

College of Water Conservancy and Environment, Zhengzhou University, Zhengzhou 450001, China;
wym311258@gs.zzu.edu.cn (Y.W.); yishengzhang@zzu.edu.cn (Y.Z.); hhriver88@gs.zzu.edu.cn (H.H.);
fenglingtong@gs.zzu.edu.cn (L.F.)

**\*** Correspondence: zhangcc@zzu.edu.cn; Tel.: +86-135-2348-0466

**Abstract:** With the accelerated urbanization process, cities are suffering from extremely heavy rain and urban storm water logging disasters in recent years. To provide reliable and effective information for urban management and emergency decision-making, the accuracy of basic data must be guaranteed in any urban rainwater model. This paper presents a novel MKFCM-MRF (Multiple Kernel Fuzzy C Means-Markov Random Field) model to segment high-resolution Unmanned Aerial Vehicle (UAV) images. The core ideas of MKFCM-MRF model are as follows. Firstly, in order to increase the correlation information between pixels, multiple-kernel functions are introduced into Fuzzy C Means (FCM) clustering algorithm, which automatically filters out the optimal weight combination among kernel functions according to the distribution characteristics of pixels in feature space. Secondly, in order to better segment the texture and edge of the image, the clustering results of multiple-kernel FCM clustering algorithm are introduced into Markov Random Field (MRF) model, a novel spatial energy function integrating fuzzy local information is constructed. Finally, based on the total of data and spatial energies, the raw clustering results are regularized by a global minimization of the energy function using its iterated conditional modes (ICM). The effectiveness of MKFCM-MRF is performed by high-resolution UAV images data. The experimental results indicate MKFCM-MRF can refine the classification map in homogeneous areas, while reducing most of the edge blurring artifact, and improving the classification accuracy compared with FCM clustering algorithm. In addition, the validation of the urban storm flood model shows that the trend of the two clustering results is the same, but the runoff producing time and the peak time of FCM clustering results are advanced, the peak flow and the total runoff are larger; the simulation results corresponding to MKFCM-MRF clustering results are more realistic.

**Keywords:** UAV; FCM; MKFCM; MRF; MKFCM-MRF

## 1. Introduction

To provide reliable and effective information for urban management and emergency decision-making, urban rainfall-flood model should first be based on ensuring the accuracy of data. However, more and more studies have recognized that in the process of urban rainfall-flood model simulation, urban buildings [1], intersections [2], and rainwater grates [3] will affect the movement of surface water in varying degrees, and small errors in topographic data will also cause changes in runoff depth, flow velocity and submergence range [4]. Therefore, the acquisition of refined urban underlying surface basic data will be the inexorable trend of urban rainwater model development. However, the composition of urban underlying surface is quite complex, which brings great challenges

to the construction of accurate and efficient urban rainwater model. With the development of remote sensing technology and remote sensing image processing technology, it is possible to quickly extract the underlying surface information of urban, such as Quick-Bird, IKONOS, SPOT, WorldView2 and other high-resolution images. Although these images can be used for target recognition, they are susceptible to cloud and fog and have a long imaging period. In recent years, Unmanned Aerial Vehicle (UAV) low-altitude image acquisition has been widely employed [5]. UAV can fly under clouds, and has the advantages of high resolution and timeliness. Therefore, UAV imaging opens up a new way to obtain the land types of underlying surface of refined urban rainwater model.

Unmanned aerial vehicle (UAV) remote sensing images collect electromagnetic wave information of various ground objects on the underlying surface. According to the difference of electromagnetic wave characteristics and spectral characteristics between different ground objects, it is possible to extract UAV image information. However, due to the complexity of electromagnetic wave characteristics and spectral characteristics of terrestrial objects, the classification of different terrestrial objects is uncertain in the process of image information extraction. Generally, remote sensing image information extraction methods are manual visual interpretation and computer automatic interpretation, which have high precision, but there are some problems such as inefficiency and boundary dislocation. Computer automatic interpretation can just make up for the above shortcomings, and the common computational automatic interpretation method is information extraction method based on the statistical characteristics of pixels. In the process of extracting, it is very important to select a suitable clustering algorithm for improving clustering accuracy. The fuzzy clustering algorithm is the most direct and effective algorithm in clustering analysis, and the FCM clustering algorithm described in References [6–9] is the most widely used. However, it still lacks in obtaining robustness to noise and outliers, especially in the absence of prior knowledge of the noise. To overcome this problem, Duan [10] proposed a partial supervision-based fuzzy c-means clustering method, and the proposed clustering method has good performance. Mehena [11] presented a spatial multiple-kernel fuzzy C-means (SMKFCM) algorithm for segmentation problem. A linear combination of multiples kernels with spatial information is used in the kernel FCM and the updating rules for the linear coefficients of the composite kernels are derived as well. By combining an evolving clustering method (ECM) with the FCM algorithm, a new method of remote sensing image segmentation is put forward. Using the FCM algorithm to solve the choice of ECM's initialization clustering centers and using the FCM to optimize the obtained centers, fuzzy clustering is completed [12]. Aiming to reduce the sensitivity, Venu [13] presented a generalized a novel MKFCM methodology with spatial information introduced as a framework for image-segmentation problem. Dhanalakshhmi [14] presented a new algorithm for a novel image-segmentation algorithm proposed which combines the Discrete wavelet transformation using "haar" function, Modified Multiple-kernel fuzzy c-means clustering (MMKFCM) and adaptive level set method. Besides building algorithm KIT2FCM to overcome some drawbacks of the conventional FCM and taking advantage of fuzzy clustering technique on the interval type 2 fuzzy set in handling uncertainty, Nguyen [15] also introduces combining the different kernels to construct the MKIT2FCM, which provides us with a new flexible vehicle to fuse different data information in the classification problems. In addition, Nguyen [16] built a composite kernel by mapping each input feature onto individual kernel space and linearly combining these kernels with the optimized weights of the corresponding kernel. Because everything is correlated, the similar things are more closely related, but the fuzzy theory does not consider the correlation of image neighborhood, and the Markov Random Field (MRF) theory points out that under the condition that the state of any pixel is known, the probability of the state of random field at the pixel is related to the state of its neighborhood, which can effectively divide the texture and edge of the image. Therefore, the new algorithm combines FCM and MRF together to computed image segmentation [17–23]. In addition, Zhou [24] developed a novel classification optimization approach integrating class adaptive MRF and fuzzy local information (CAMRF-FLI) for high spatial resolution multispectral imagery (HSRMI). Binu [25] presented a hybrid algorithm, called MKF-Cuckoo which is the hybridization of cuckoo search algorithm with the multiple-kernel-based fuzzy c means algorithm.

The results indicate it can refine the classification map in homogeneous areas, while reducing most of the edge blurring artifacts, and improving the classification accuracy compared with some conventional approaches. In view of this, this paper constructs a MKFCM clustering algorithm based on the FCM clustering algorithm, and the clustering results of MKFCM algorithm are introduced into an MRF model, with the aim of constructing a novel clustering algorithm based on Multiple Kernel Fuzzy C Means and Markov Random Field model (MKFCM-MRF).

## 2. Acquisition and Preprocessing of UAV Images

### 2.1. Acquisition of UAV Images

The underlying surface and rainwater pipe network of Zhengzhou University have their own independent rainwater drainage system, which is similar to the microcosm of a city. Therefore, Zhengzhou University is chosen as the flight test area. As shown in Figure 1.

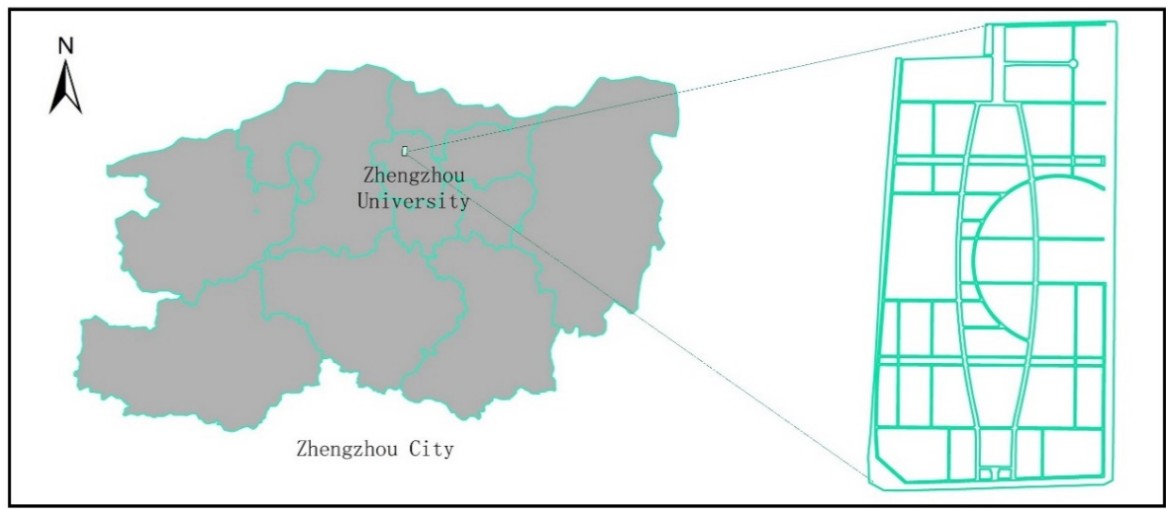

**Figure 1.** Unmanned Aerial Vehicle flight area.

At present, the common civil UAVs are rotor UAVs and fixed-wing UAVs. Fixed-wing electric mapping UAV was used in this flight, with length × wingspan of 1.1 m × 2 m, takeoff weight of 3.5 kg, range of 3 km, flight speed of 450 km/h and flight time of 150 min. The UAV was equipped with ground remote control system and SONY-a 5100. The focal length of the camera is 24 mm and the CMOS (Complementary Metal-Oxide Semiconductor) is 25.3 mm × 15.6 mm. The area of Zhengzhou University is 2.12 km$^2$: about 2078 m long from south to north and 1052 m wide from east to west. This flight consisted of 9 routes, each of which was about 2567 m long. The overlap degree of course is 70% and the side overlap degree is 55%. The flight time was noon on 17 July 2017. The weather was very good and cloudless. The flying height was 306.38 m; Ground Sampling Distance was 0.05 m.

### 2.2. Preprocessing of UAV Images

Before classifying the underlying surface features in the study area, it was necessary to use UAV data processing software to ortho-rectify the image and obtain the Digital Orthophoto Map (DOM). At present, there are many data processing software for UAV, such as PIXELGRID, DPGRID, IPS, HOTOMOD, GodWork, Cloud-AT, MAP-AT, SOCETSET and Pix4Dmapper (Upgraded version of Pix4UAV). Compared with the above software, Pix4Dmapper uses one-key processing mode. It encapsulates the algorithms of empty-three calculation, adjustment calculation, ortho-correction and mosaic, provides a unified data input interface, and outputs the results after background calculation. Its operation needs neither professional knowledge nor manual intervention, and its automation degree

and calculation accuracy are high. Therefore, Pix4Dmapper was selected as the automatic and fast UAV data processing software.

The Pix4Dmapper image processing flow is shown in Figure 2. First, data such as image data, POS (Position and Orientation System) data and control point data is prepared. To ensure that the photo number of image data corresponds to the photo number of POS data, aerial photographs containing GPS coordinates are sorted out and POS data files are checked at the same time. Second, the project management is opened and the privileges validated, new project built and photos added. The software will automatically read the attribute information of photos, such as image coordinate system, geographic location and direction, camera model, photo grouping and other parameters; after selecting the output coordinate system and processing template, the photos will be processed, and in the process of image processing (a) initialization processing, (b) point cloud and texture, (c) DSM, DOM and index are important processing steps, and DTM, contours and three-dimensional grids can be generated on the basis of the previous processing. Finally, the export results, that is, after image processing, can export quality reports and project subfolders (project subfolders include the output results of index, DSM (Digital Surface Model), DOM (Digital Orthophoto Map), and point cloud files). The DOM of UAV image in this research area is shown in Figure 3 (the geo-referencing of DOM only uses the GPS tag from the used UAV).

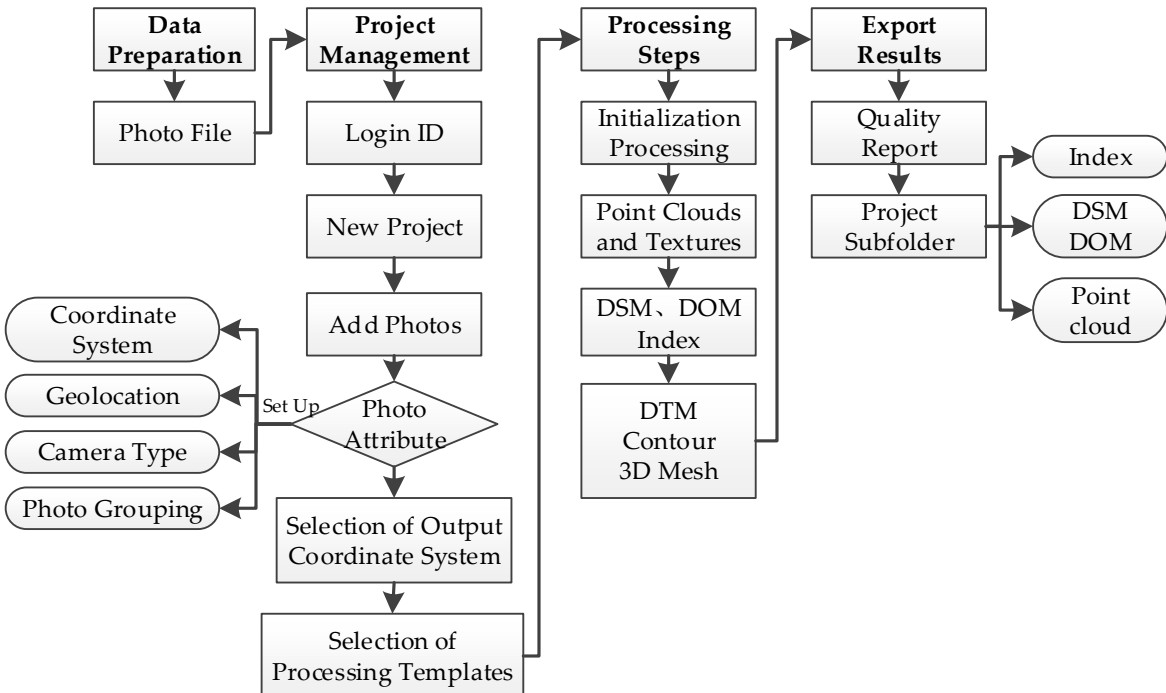

**Figure 2.** Pix4Dmapper Image Processing Flow Chart (DSM: Digital Surface Model. DOM: Digital Orthophoto Map).

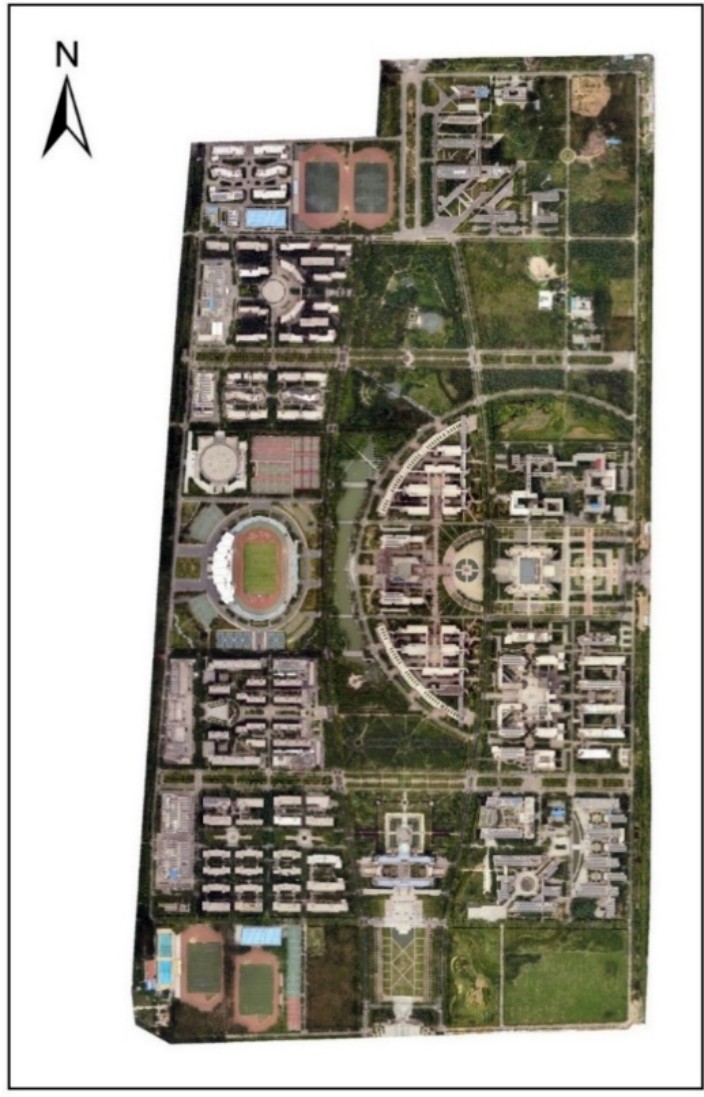

**Figure 3.** DOM of UAV (Unmanned Aerial Vehicle) Images in Research Area.

## 3. Methods

### 3.1. Multiple-Kernel Fuzzy C-Means Clustering Algorithm

FCM clustering algorithm—a generalization of the hard c-means algorithm—yields extremely good results in image region clustering and object classification [13]. Given a data set $X = \{x_1, x_2, \ldots, x_n\}$, where the data point $x_j \in \Xi \subseteq R^p (j = 1, 2, \ldots, n)$, $n$ is the number of data, and p is the input dimension of a data point, FCM groups $X$ into c clusters by minimizing the weighted sum of distances between the data and the cluster centers or prototypes defined as:

$$Q = \sum_{i=1}^{c} \sum_{j=1}^{n} u_{ij}^m \|x_j - o_i\|^2 \tag{1}$$

Here, $\|\cdot\|$ is the Euclidean, $u_{ij}$ is the membership of data $x_j$ belonging to cluster $i$, which is represented by the prototype $o_i$. The constraint on $u_{ij}$ is $\sum_{i=1}^{c} u_{ij} = 1$, and $m$ is the fuzzification coefficient ($m = 2$).

The learning algorithm of FCM iteratively updates $u_{ij}$ as

$$u_{ij} = \frac{\|x_j - o_i\|^{-1/(m-1)}}{\sum_{i=1}^{n} \|x_j - o_i\|^{-1/(m-1)}} \tag{2}$$

Update the cluster centers:

$$o_i = \frac{\sum_{j=1}^{n} u_{ij}{}^m x_j}{\sum_{j=1}^{n} u_{ij}{}^m} \tag{3}$$

MKFCM is a clustering algorithm based on multi-kernel function. Because the Gauss function belongs to the robust radial basis function (RBF), the RBF has a good anti-noise effect on the data, and the range of the value of the Gauss kernel function is between 0 and 1, which effectively simplifies the calculation process, so the Gauss function is chosen as the kernel function. Here, the new composite kernel $K_L$ is defined as:

$$k_L = w_1^m k_1 + w_2^m k_2 + \cdots + w_l^m k_l \tag{4}$$

where the regulation on weights, $w_1$, $w_2$, ... $w_l$, is $\sum_{i=1}^{l} w_l = 1$. $k_1$ is pixel intensities of the Gaussian kernel: $k_1(x_i, x_j) = \exp\left(-r|x_i - x_j|^2\right)$; $k_2$ is spatial information of the Gaussian kernel: $k_2(x_i, x_j) = \exp\left(-r|x_i' - x_j'|^2\right)$; $k_3$ is the Gaussian kernel for texture information: $k_3(x_i, x_j) = \exp\left(-r\left|[x_i' - s_i] - [x_j' - s_j]\right|^2\right)$. If we set $x_j$ in Equation (2) as $x_i = [x_j, x_i', s_j] \in R^3$ where $x_j$ is intensity of pixel $j$, $x_i'$ is filtered intensity of pixel $j$, $x_j'$ is filtered intensity of pixel $i$, $s_i$ is standard variance of the intensity of pixels in the neighbourhood of pixel $i$, $s_j$ is standard variance of the intensity of pixels in the neighbourhood of pixel $j$.

Then the objective function of the MKFCM with the linearly combined kernel is still the weighted sum of distances between the data and prototypes in the kernel space:

$$Q = \sum_{i=1}^{c} \sum_{j=1}^{n} u_{ij}^m \|\varphi_L(x_j) - o_i\|^2 \tag{5}$$

where, $\varphi_L$ is the transformation derived from the linearly combined kernel $k_L(x_i, x_j) = <\varphi_L(x), \varphi_L(y)>$ [$k_L$ is defined in Equation (4)]. The learning algorithm of MKFCM-K iteratively updates $u_{ij}$ as:

$$u_{ij} = 1 / \sum_{k=1}^{c} \left(d_{ij}^2 / d_{kj}^2\right)^{\frac{1}{m-1}} \tag{6}$$

where

$$d_{ij}^2 = k_L(x_j, x_j) - \frac{2 \sum_{h=1}^{n} u_{ih}^m k_L(x_h, x_j)}{\sum_{h=1}^{n} u_{ih}^m} + \frac{\sum_{h=1}^{n} \sum_{l=1}^{n} u_{ih}^m u_{il}^m k_L(x_h, x_j)}{\left(\sum_{h=1}^{n} u_{ih}^m\right)^2} \tag{7}$$

By introducing the Lagrange term of the constraint of weights $w_i (i = 1, 2, \ldots, l)$ into the objective function, and the updated formula is as follows:

$$Q = \sum_{i=1}^{c} \sum_{j=1}^{n} u_{ij}^m \varphi_L(x_j) - o_i{}^2 + \eta(1 - \sum_{i=1}^{l} w_i) \tag{8}$$

By taking derivative of $Q$ over $w_i$ and assuming the results to zero, and we obtain the updating rule of the weights $w_i (i = 1, 2, \ldots, l)$.

$$\frac{\partial Q}{\partial w_i} = 0 (i = 1, 2, \ldots, l) \tag{9}$$

Yield to:

$$w_i = \frac{1}{\sum_{h=1}^{n} \left(\frac{Q_i}{Q_h}\right)^{\frac{1}{(b-1)}}} \tag{10}$$

where:

$$Q_h = \sum_{i=1}^{c} \sum_{j=1}^{n} u_{ij}^{m} \|\varphi_h(x_j) - o_i\|^2 (h = 1, 2, \ldots, l) \tag{11}$$

Here, $\varphi_h$ is the transform function defined by $k_h(h = 1, 2, \ldots, l)$ in Equation (1) and

$$\|\varphi_h(x_j) - o_i\|^2 = k(x_j, x_j) - \frac{2 \sum_{l=1}^{n} u_{il}^{m} k_h(x_l, x_j)}{\sum_{l=1}^{n} u_{il}^{m}} + \frac{\sum_{g=1}^{n} \sum_{l=1}^{n} u_{ig}^{m} u_{il}^{m} k_h(x_g, x_l)}{(\sum_{g=1}^{n} u_{ig}^{m})^2} \tag{12}$$

In the selection of kernel function, this paper uses Gauss function to carry out experiments. This function belongs to the robust radial basis function, which has a good anti-noise effect on data, and the range of value of Gauss kernel function is between 0 and 1, which effectively simplifies the calculation process.

*3.2. Markov Random Field Theory*

MRF is a modeling method based on conditional probability theory to describe spatial sequence correlation. MRF can effectively divide the texture and edge of the image, and has strong spatial constraints and few model parameters, so it is widely used.

In MRF, the gray-scale array of UAV image pixels is called observation sequence, and the classification information of each pixel is called labeling sequence, and each component of the labeling sequence is independent of each other. That is to say, for every pixel point in UAV image, if the probability function of the *i*-th pixel point belongs to the *k*-th object is expressed as $P(y_i^k | X_i = k)$, abbreviated as $P(y_i | x_i)$; The prior probability of the system in the field of the location of the *i*-th pixel is expressed as $P(X_i = k | X_{Ni})$, which is abbreviated as $P(x_i | x_{Ni})$. The posterior probability is expressed as $P(X_i = k | y_i^k)$ and abbreviated as $P(x_i | y_i)$. Then the image clustering problem can be approximated as follows:

$$P(x_i | y_i) \propto argmax\{P(y_i | x_i) \cdot P(x_i | x_{Ni})\} \tag{13}$$

According to the Hammersley–Clifford theorem, MRF random field is equivalent to Gibbs random field, and the prior probability and posterior probability distribution in MRF neighborhood system obey Gibbs distribution, that is:

$$\begin{cases} P(x_i | x_{Ni}) = \frac{1}{Z} exp\left[-\frac{U(x_i | x_{Ni})}{T}\right] \\ P(x_i | y_i) = \frac{1}{Z} exp\left[-\frac{U(x_i | y_i)}{T}\right] \end{cases} \tag{14}$$

where $Z$ is a normalized function, $U(x_i | x_{Ni})$ is a priori energy function for image clustering marking problem, $U(x_i | y_i)$ is a posterior energy function, and $T$ is a temperature constant.

Formula (14) can be further simplified as follows:

$$\begin{cases} P(x_i | x_{Ni}) \propto e^{-U(x_i | x_{Ni})} \\ P(x_i | y_i) \propto e^{-U(x_i | y_i)} \end{cases} \tag{15}$$

Formula (15) is substituted into Formula (13) and logarithms are taken on both sides to obtain:

$$ln\left[e^{-U(x_i | y_i)}\right] \propto argmax\left\{ln\left[e^{-U(y_i | x_i)}\right] \cdot ln\left[e^{-U(x_i | x_{Ni})}\right]\right\} \tag{16}$$

The minimum posteriori energy function $U(x_i|y_i)$ is used to equivalent the maximum posteriori probability $P(x_i|y_i)$. The results show that:

$$U(x_i|y_i) \propto argmax\{U(y_i|x_i) + U(x_i|x_{Ni})\} \tag{17}$$

The above formula can be abbreviated as:

$$U(x_{Fi}|y_i) = U(y_i|x_i) + U(x_i|x_{Ni}) \tag{18}$$

In this paper, iterative conditional mode (ICM) algorithm is used to infer MRF. It assumes that the label field of the image is I, except for the $i$-th pixel, the label field is $x_{I-i}^{\{k\}}$, and K is the number of iterations. By continuously updating each pixel label $x_i^{\{k\}}$ to $x_i^{\{k+1\}}$, ICM algorithm makes the posterior energy function $U(y_{Fi}|x_i)$ get the minimum value, and then the final clustering result is obtained. The flow chart of ICM algorithm is shown in Figure 4.

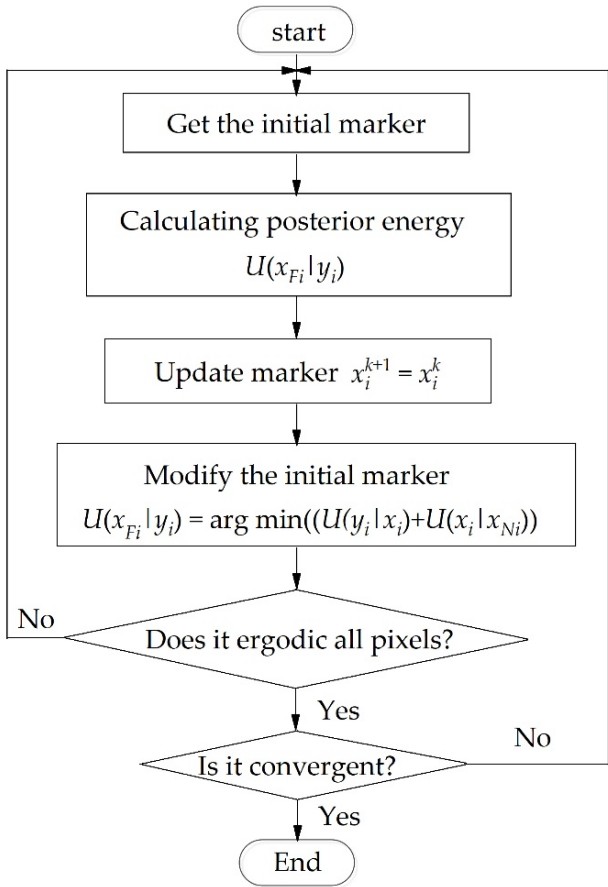

**Figure 4.** Flow chart of iterative conditional mode (ICM) algorithm.

### 3.3. Constructing MRF-MKFCM Model Based on MKFCM and MRF

MKFCM clustering algorithm maps the original data to a new feature space through a non-linear mapping relationship of multi-core functions, which can not only increase the linear separability of data, but also describe the various features of data with different kernel functions; moreover, according to the characteristics of data distribution, the optimal weight can be automatically selected, and the combination of various kernel functions can be carried out. MRF can effectively divide the texture and edge of the image and has strong spatial constraints and fewer model parameters. Therefore, this paper introduces the clustering results of MKFCM algorithm into MRF model. Its construction method is as follows:

Firstly, based on MKFCM clustering, the prior probability of each pixel is obtained to construct the likelihood function energy of the *i*-th pixel in the object category of the observation site:

$$U(y_i|x_i) = -ln(\sum_{j=1}^{c} \mu_{ij}) \tag{19}$$

Here, *c* denotes the number of types of objects.

Secondly, the clustering of central pixels is constrained by the prior clustering information of neighborhood pixels, and the local spatial correlation information is integrated into the clustering. That is to say, the prior probability energy corresponding to the *i*-th pixel in the label field is obtained:

$$U(x_i|x_{Ni}) = -\ln\left[\sum_{j\in Ni} \exp\left(-\frac{\|x_i - x_j\|^2}{2\sigma^2}\right)\right] \tag{20}$$

Here, $\sigma^2$ is a normalized term, the range of $U(x_i|x_{Ni})$ is limited between 0 and 1.

## 4. Results

To improve the accuracy of urban rainfall runoff model simulation, it is necessary to accurately identify the types and distribution of underlying surface. In this experiment, the image data of Unmanned Aerial Vehicle (UAV) on the underlying surface of Zhengzhou University are pretreated in Section 2.2 of this chapter, and the FCM clustering algorithm and MKFCM clustering algorithm were modeled by MATLAB. First, the eigenvalues of pixels in UAV images were extracted and the eigenvalue matrix is constructed. Second, the clustering category was determined according to the actual distribution of ground objects. In this paper, the underlying surface features of Zhengzhou University are divided into five categories: grassland, woodland, buildings, roads, water bodies, etc. Then, the parameters of the model were optimized, and the gradient descent method was used to select the relevant parameters of the FCM model and MKFCM model, and the set of optimized parameters was obtained. Finally, for the MKFCM model, the kernel function and its parameters were selected, and the optimal parameter combination among the kernel functions was automatically selected by the distribution characteristics of the pixel eigenvalues. For MKFCM-MRF model, the results of MKFCM model were introduced into MRF, and then ICM algorithm was used to infer MRF in order to get the final clustering results. For comparison, the clustering results of FCM and MKFCM-MRF models are presented, as shown in Figures 5 and 6.

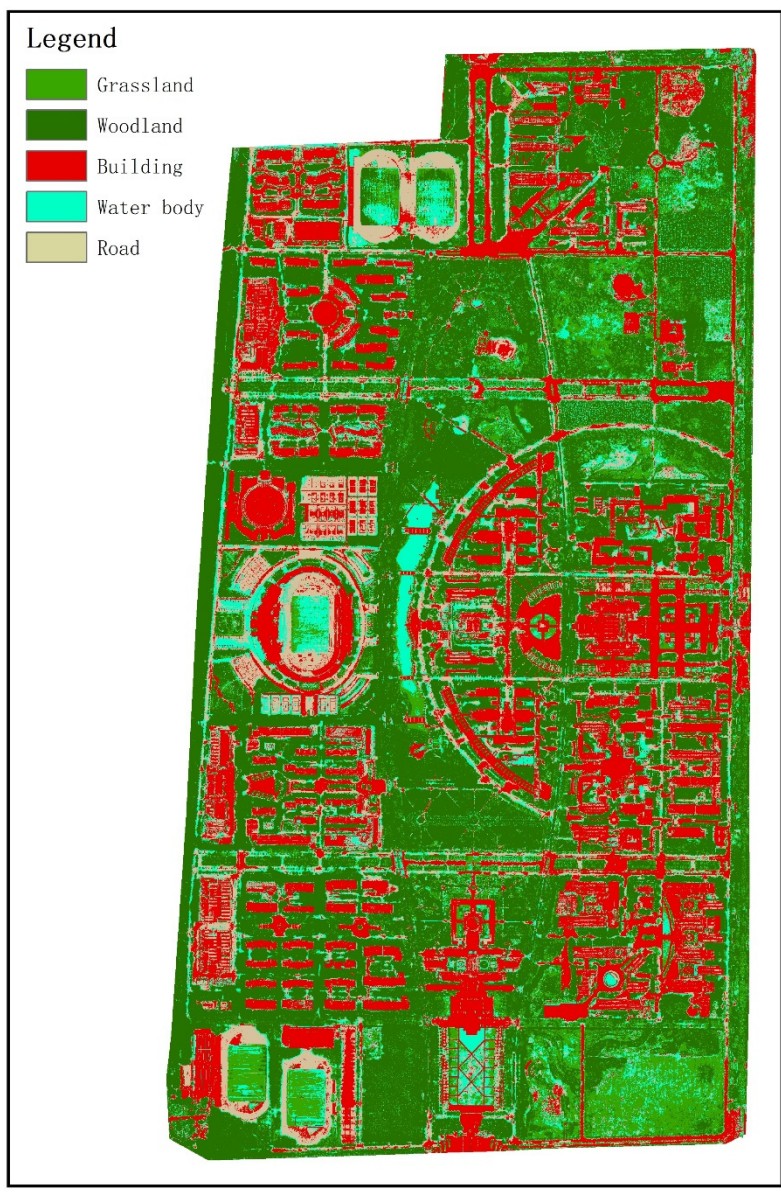

**Figure 5.** Fuzzy C Means (FCM) model clustering results.

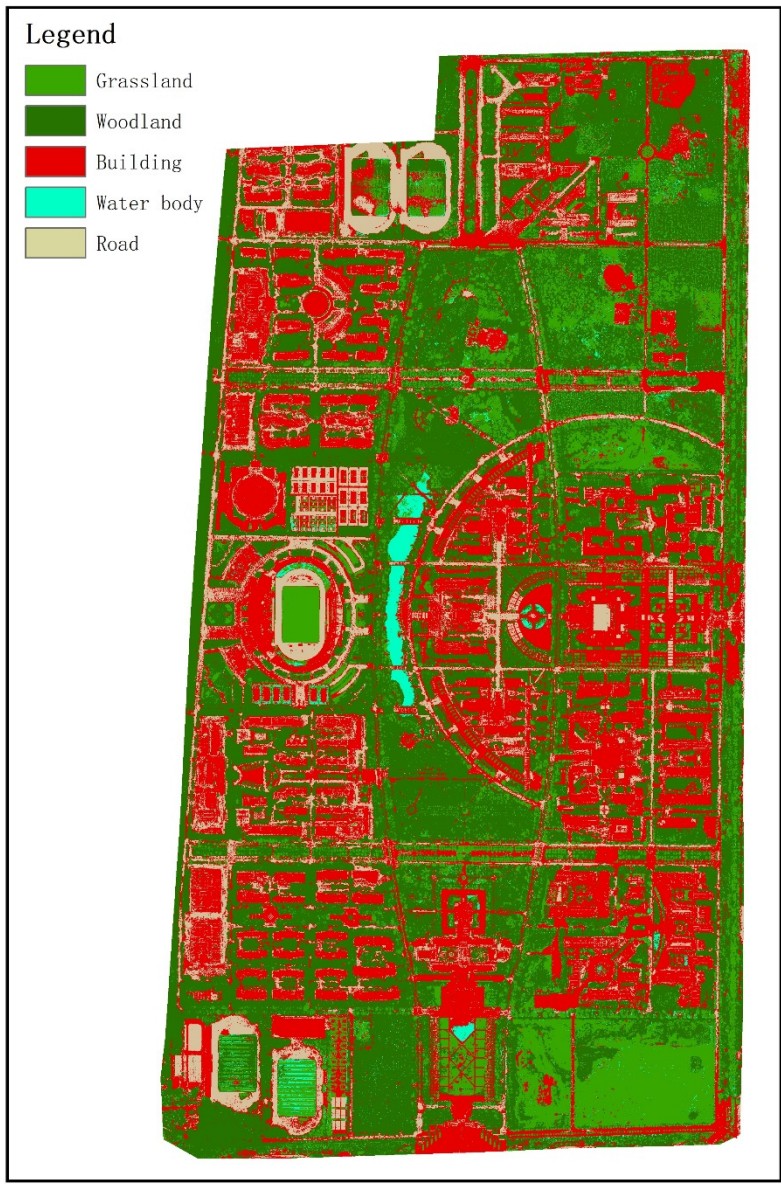

**Figure 6.** Multiple Kernel Fuzzy C Means - Markov Random Field (MKFCM-MRF) model clustering results.

When constructing the runoff generation and concentration model of urban rainwater and flood underlying surface, the underlying surface is usually divided into permeable lands (such as grassland, woodland, etc.), impermeable lands (such as buildings, roads, etc.), and water bodies (such as open channels, reservoirs, etc.) [26]. Among them, the infiltration of different land types in the permeable lands is different, which has a greater impact on the urban rainfall and flood surface runoff. The impermeable land not only provides important parameters for the urban storm runoff model [27,28], but also affects the temperature, evapotranspiration and soil water content of the city [29,30]. Water body refers to the impermeable land with stagnant storage capacity, usually without runoff. Therefore, it is necessary to re-integrate the clustering results of FCM and MKFCM-MRF models above to get the underlying surface types needed to construct the urban rainfall runoff model (as shown in Figures 7 and 8).

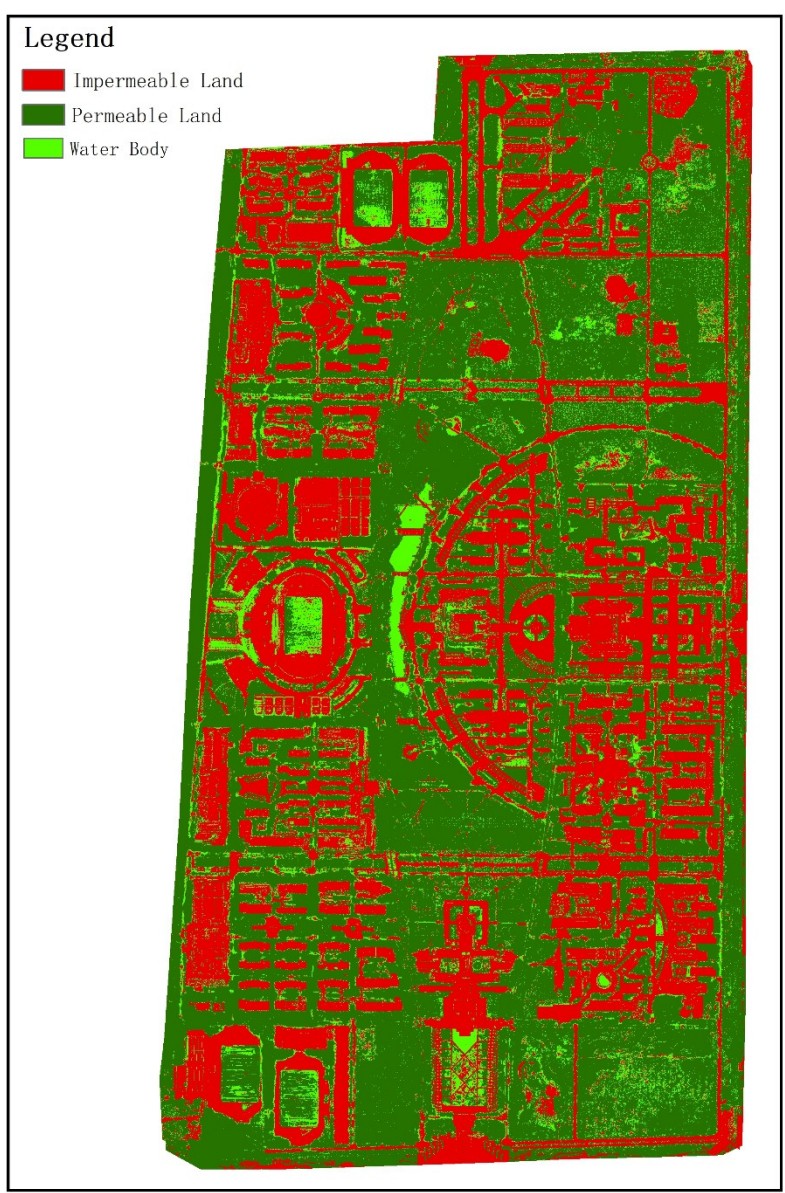

**Figure 7.** Fuzzy C Means (FCM) model clustering results.

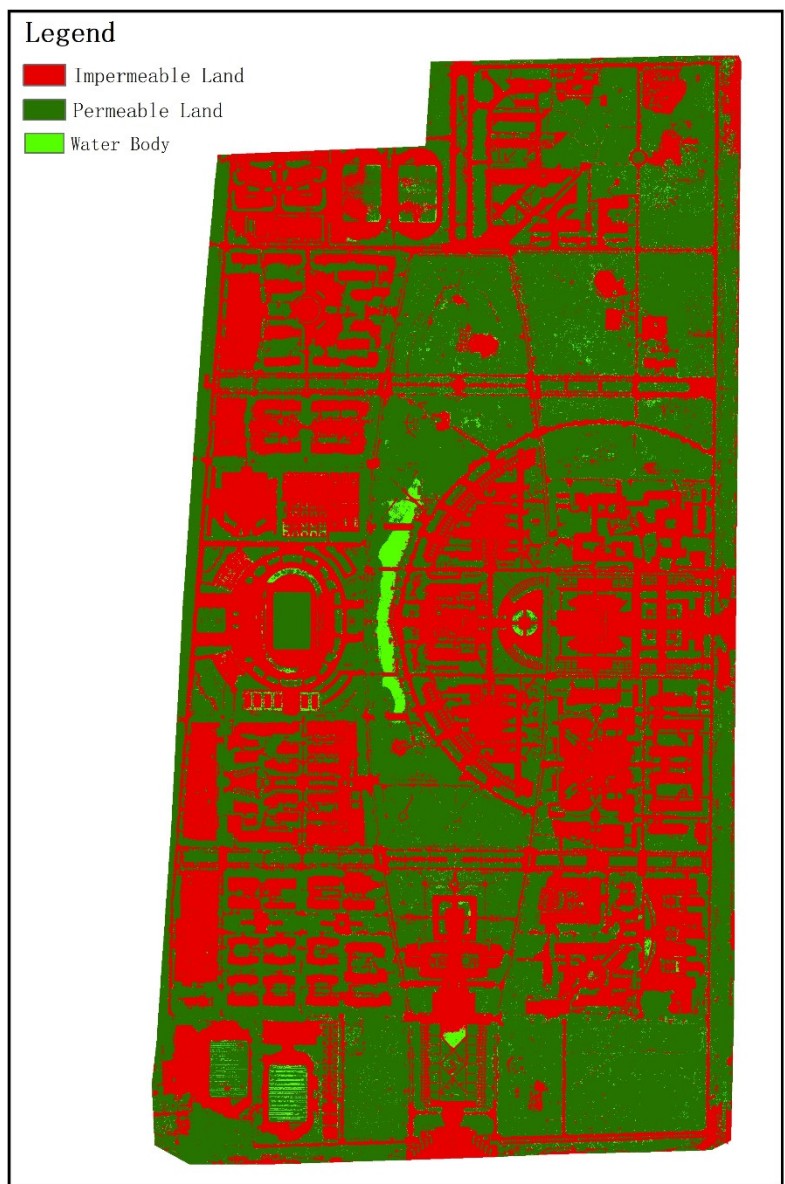

**Figure 8.** Multiple Kernel Fuzzy C Means - Markov Random Field (MKFCM-MRF) model clustering results.

## 5. Discussion

### 5.1. Accuracy Analysis

Accuracy analysis can effectively evaluate the quality of clustering results, so accuracy evaluation is an indispensable part of image clustering process. Because the spatial resolution of UAV image in this research area is as high as 0.05 m × 0.05 m, the number of pixels is 22,355 × 42,098, and the amount of data is too large, the accuracy of some areas in the research area is analyzed. The selected range is shown in Figure 9. There are 1,506,077 validation pixels, including 509,559 pixels of water bodies, 285,566 pixels of impermeable lands and 7,100,862 pixels of permeable lands.

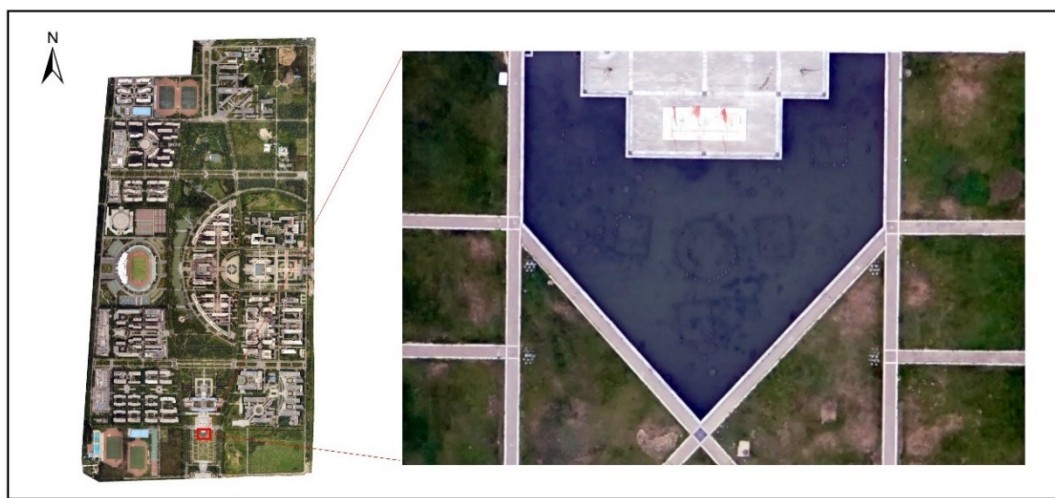

**Figure 9.** Areal map selected for accuracy analysis.

Confusion matrix is a standard format for accuracy evaluation. It can visually show the correct rate of different clustering categories and the situation that errors are clustered into other categories. Specific evaluation indicators include user accuracy, producer accuracy, overall accuracy and Kappa coefficient. These accuracy indicators reflect the accuracy of image clustering from different aspects. User accuracy represents the probability that the pixels gathered on the image to represent a certain kind of objects on the ground actually represent that kind of objects. Producer precision represents the probability that reference pixels are clustered correctly. The overall accuracy represents the probability that the clustering result of a randomly selected pixel sample is consistent with the real object type. Kappa coefficients can reflect the overall clustering accuracy more comprehensively by taking into account all the factors in the confusion matrix. In this paper, when evaluating the clustering accuracy of FCM and MKFCM-MRF models, the selected visual interpreted images are taken as the theoretical truth images of underlying surface feature information to establish the confusion matrix, as shown in Tables 1 and 2.

**Table 1.** Accuracy Assessment of Unmanned Aerial Vehicle (UAV) Image Clustering for FCM.

| Image Elements | | Reality Classes | | | Total Count | User's Accuracy (%) |
|---|---|---|---|---|---|---|
| | | Water Bodies | Impermeable Lands | Permeable Lands | | |
| Clustering Classes | Water Bodies | 485,128 | 19 | 243,691 | 7,288,338 | 66.56 |
| | Impermeable Lands | 6035 | 285,637 | 90,308 | 381,980 | 74.78 |
| | Permeable Lands | 18,396 | 0 | 376,863 | 395,259 | 95.35 |
| Total Count | | 509,559 | 285,656 | 710,862 | 1,506,077 | |
| Product's Accuracy (%) | | 95.21 | 99.99 | 53.01 | | |
| Overall Accuracy = 0.76; Kappa Coefficient = 0.64 | | | | | | |

**Table 2.** Accuracy Assessment of Unmanned Aerial Vehicle (UAV) Image Clustering for MRF-MKFCM.

| Image Elements | | Reality Classes | | | Total Count | User's Accuracy (%) |
|---|---|---|---|---|---|---|
| | | Water Bodies | Impermeable Lands | Permeable Lands | | |
| Clustering Classes | Water Bodies | 356,184 | 0 | 3813 | 359,997 | 98.94 |
| | Impermeable Lands | 149,973 | 285,297 | 39,132 | 474,402 | 60.13 |
| | Permeable Lands | 3402 | 359 | 667,917 | 671,678 | 99.44 |
| Total Count | | 509,559 | 285,656 | 710,862 | 1,506,077 | |
| Product's Accuracy (%) | | 69.9 | 99.87 | 93.95 | | |
| Overall Accuracy = 0.87; Kappa Coefficient = 0.80 | | | | | | |

In remote sensing images, water and building shadows and vegetation shadows have similar spectral characteristics, and roads and buildings have similar spectral characteristics, so that the possibility of misclassification between them is relatively high. Because roads and buildings belong to impermeable lands, the misclassification of roads and buildings will not affect the accuracy evaluation of UAV image clustering. This paper focuses on the clustering results between water and building shadows and vegetation shadows, that is, the clustering results between water body, impermeable lands and permeable lands, and then analyses the changes of the accuracy evaluation indices of the three lands.

According to Tables 1 and 2, the FCM model clustered impermeable lands well. Permeable lands were divided into water bodies accurately, and a small number of water bodies were divided into permeable lands by mistake. The MKFCM-MRF model clustered impermeable lands well. Water bodies were divided into impermeable lands accurately, and a small number of permeable lands were divided into impermeable lands by mistake. Comparing the user accuracy and producer accuracy of the two models, we can see that the MKFCM-MRF model has 14.65% lower user accuracy than the FCM model in impermeable lands and 25.31% lower producer accuracy than the FCM model in water bodies. The other clustering accuracy evaluation indices are greater than that of the FCM model. The producer accuracy of the permeable lands is improved by 40.94% and the user accuracy of water bodies are increased by 32.38%. By comparing the overall accuracy and Kappa coefficient of the two models, we can see that the MKFCM-MRF model is much higher than the FCM model, in which the overall accuracy is improved by 0.11, and the Kappa coefficient is increased by 0.16. The above results are all because FCM only considers the feature vectors of the pixels in clustering, ignores the spatial relationship between adjacent pixels, and cannot distinguish different objects with similar spectral characteristics. The MKFCM algorithm in MKFCM-MRF model can map the original feature vectors into the high-dimensional feature space through the kernel function, and automatically optimize the combination of the kernel functions according to the distribution characteristics of the feature vectors in the space, so as to improve the linear separability between different types of data and distinguish the heterogeneous objects with similar spectral characteristics. There are discrete holes and cluttered points in FCM clustering results, while MKFCM-MRF clustering results show that the noise is greatly reduced and the boundary is smoother. This is because the Markov Random Field model based on fuzzy theory can not only deal with the randomness of image clustering process, but also deal with its fuzziness, without losing the spatial information of the image.

## 5.2. Verification of Urban Storm Flood Model

Since the 1960s, hydrological models and hydrodynamic models of urban storm floods have emerged endlessly. Many scientific research institutes and scholars in the United States [31,32], Britain [33,34], Denmark [35], Australia [36] and China [37] have developed many kinds of urban storm flood models from the perspectives of simple concepts to complex hydrodynamics, or from statistics to certainty, such as TRRL(Transport and Road Research Laboratory) flood model, ILLUDAS, SWMM(Storm Water Management Model), UCURM, STORM, Wallingford, XP-SWMM, MIKE FLOOD, IFMS/Urban and so on, among which the SWMM is a widely used program for simulating urban runoff quantity and quality. Therefore, SWMM is selected as the urban flood model to validate the accuracy of land cover types in UAV images using MRF and MKFCM clustering techniques. The flight area of UAV is taken as the research object, which has 773 underlying sub catchment areas, and these sub catchment areas are connected with the rainwater pipe network. That is to say, urban storm flood has experienced three links: the urban runoff yield, the overland flow, and storm and sewer flow, as shown in Figure 10.

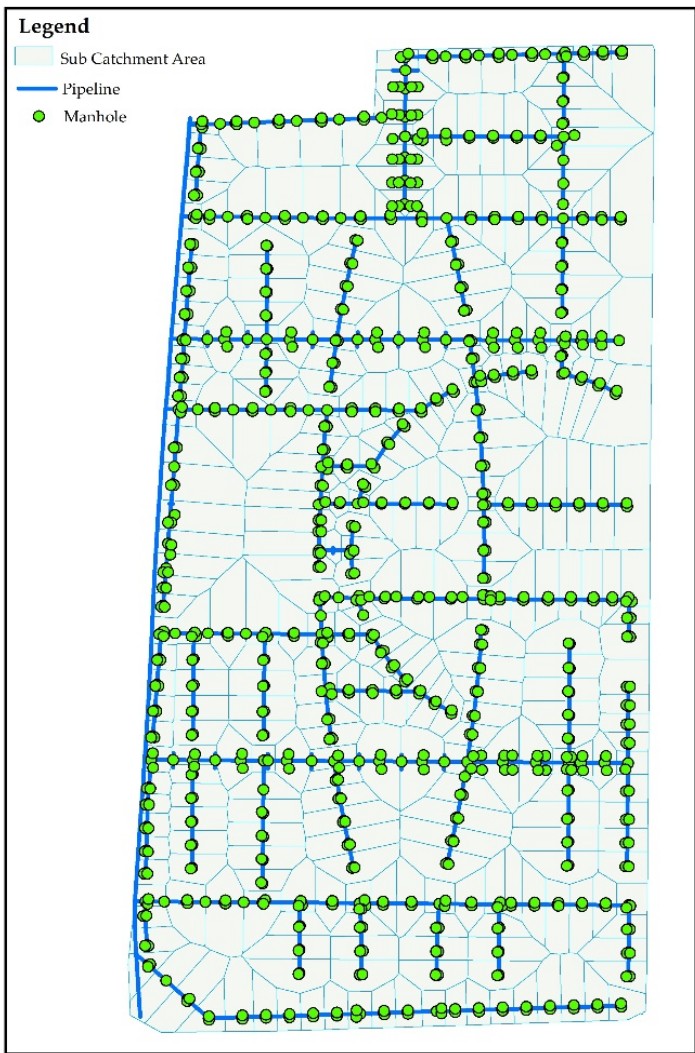

**Figure 10.** Access to storm water drainage.

The land cover types obtained by FCM and MKFCM-MRF clustering are used as the basic data for the calculation of urban storm flood model, and two rainfall events with different intensity and duration are selected as the water boundary conditions. At the same time, the Manning roughness coefficient of impervious area is 0.015 and that of permeable area is 0.24. The maximum infiltration rate is 75.25 mm/h, the minimum infiltration rate is 3.5 mm/h and the decline rate of infiltration is 3. In addition, because the impermeability is calculated according to the land cover types in each sub catchment area, that is, different land cover types give different impermeability to the sub catchment area, which stipulates that the impermeability of buildings and roads in the impermeable area is 100% and 90%, the impermeability of grasslands and trees in the permeable area is 20% and 40%, and the impermeability of water body is 0. The flow duration curve of the outlet under different clustering algorithms and rainfall conditions in the study area are shown in Figure 11 and the simulation results of SWMM are shown in Table 3.

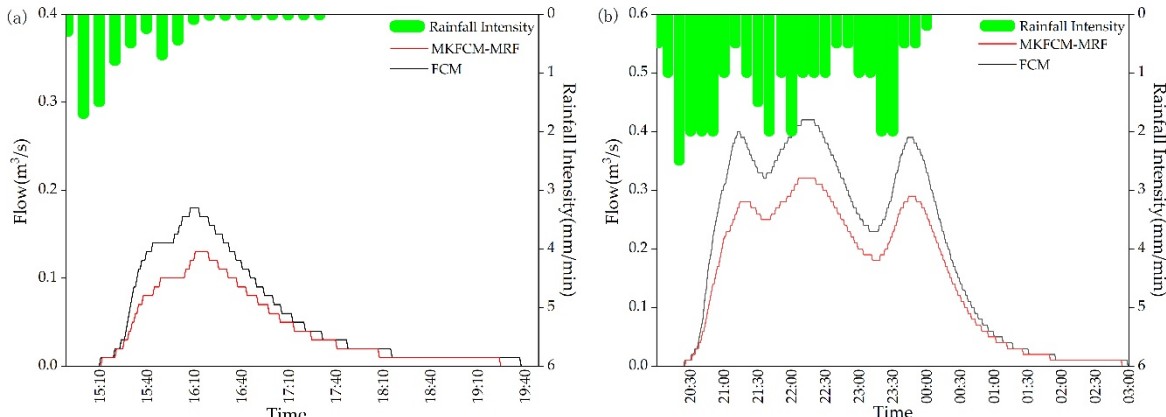

**Figure 11.** Comparing the flow duration curve of outlet with different clustering algorithms. (**a**) Small and Medium Rainfall of 12 August 2017; (**b**) Extremely heavy rain of 1 August 2018.

**Table 3.** Comparing the simulation results of Storm Water Management Model (SWMM) under different clustering algorithms.

| Rainfall Date | Clustering Algorithm | Runoff Producing Time | Peak Time | Peak Flow $(m^3/s)$ | Total Runoff $(m^3)$ |
|---|---|---|---|---|---|
| 12 August 2017 | FCM | 15:12 | 16:12 | 0.18 | 978.946 |
| | MKFCM-MRF | 15:13 | 16:13 | 0.13 | 734.464 |
| 1 August 2018 | FCM | 20:21 | 22:15 | 0.42 | 4703.723 |
| | MKFCM-MRF | 20:22 | 22:16 | 0.32 | 3591.477 |

As can be seen from Figure 9 and Table 3, the SWMM simulation results of the MKFCM-MRF clustering algorithm and FCM clustering algorithm under different rainfall conditions (left figure: 12 August 2017; right figure: 24 July 2018) show that the change trend of flow duration curve is consistent and changes with the change of rainfall intensity. This is due to the same distribution of rainwater pipe network, the same number of rainwater wells, the same calculation principle, and the same set of parameters in SWMM model. Although different types of ground objects extracted by different clustering algorithms are used as the basic data of the model in the study area, they do not affect the trend of the flow duration curve of the outlet. Comparing the simulation results of SWMM under MKFCM-MRF and FCM clustering algorithm, we can see that the runoff producing time and the peak time of MKFCM-MRF algorithm are slightly delayed, and the peak flow and total runoff are also smaller. This is because under the same conditions of rainfall, sub catchment division and runoff path, impervious areas (roads) under FCM clustering algorithm is misclassified into pervious areas, which increases the area of surface runoff, while the water bodies in MKFCM-MRF clustering are misclassified as impervious areas, however, because the area of water body in the whole study area is very small, so the overall impact is not large.

## 6. Conclusions

A novel MKFCM-MRF model for UAV image clustering has been presented in this paper. The proposed model is effective in refining clustering imaging in homogenous areas and simultaneously preserving more details. The advantages of introducing MKFCM algorithm and MRF algorithm are: (1) the original eigenvector can be mapped to high-dimensional feature space by using the kernel function, and the kernel function can be automatically optimized according to the distribution characteristics of the eigenvector in space, so as to improve the linear separability between different types of data, so as to distinguish the similar spectral features of the terrain. (2) In addition to noise reduction, edge information is well preserved, which can provide a good trade-off between over smooth and spatial regularization. Through the accuracy analysis of clustering results of Unmanned

Aerial Vehicle images based on FCM and MKFCM-MRF models, we can see that MRF-MKFCM model has better clustering accuracy of UAV images. The urban storm flood model simulation results of MKFCM-MRF clustering algorithm and FCM clustering algorithm show that the change trend of flow duration curve is consistent and changes with the change of rainfall intensity. However, because the impervious areas are misclassified into pervious areas in FCM clustering results, the runoff producing time and the peak time of FCM clustering results are advanced, the peak flow is on the high side, and the total runoff is also on the high side. While the water bodies in MKFCM-MRF clustering are misclassified as impervious areas, because the area of water body in the whole study area is very small, the overall impact is not large. Hence, the MKFCM-MRF model is an effective optimization approach for UAV image clustering. In subsequent studies, more research will be carried out on methods of optimizing the energy function, and the proposed algorithm will be tested on other images.

**Author Contributions:** Data curation, Yanmei Wang and Lingtong Feng; Formal analysis, Yanmei Wang and He Huang; Funding acquisition, Chengcai Zhang; Methodology, Yanmei Wang; Software, Yisheng Zhang; Writing—review & editing, Yanmei Wang and Lingtong Feng.

**Funding:** This research was supported by the Major Project National Natural Science Foundation of China (NO. 51739009), the Scientific and Technological Research Program of Henan Province (NO. 182102210017), for which the authors are grateful.

**Conflicts of Interest:** The authors declare no conflict of interest.

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
