# Peer review of "Obtaining Land Cover Type for Urban Storm Flood Model in UAV Images Using MRF and MKFCM Clustering Techniques"

_ijgi, doi:10.3390/ijgi8050205_

Round 1

Reviewer 1 Report

I found the paper to be interesting, but the contribution is not made clear enough in the Introduction part. It is not sufficiently detailed. The novelty, merit and/or contribution of this paper should be clearly shown in Introduction part. 

My main concern is that novelties with respect to existing strategies seem limited. Results are not convincing. They could probably be improved.

The interest of this research must be actually pointed out in the conclusion and demostrated in the result section.

Some specific concerns are as follows:

Line 121-134 --> ; Secondly.....error; Thirdly......method; Thirdly.........

Line 139-140 --> 

FCM clustering algorithm a generalization of the hard c-means algorithm yields extremely

good results in image region clustering and object classification.??

Line 278-279

Please check the number of pixels

Tables...  It would be interesting to know the value of Global Accuracy

The research and the manuscript must be improved by authors

Author Response

Dear reviewer:

On behalf of my co-authors, we thank you very much for giving us an opportunity to revise our manuscript, we appreciate you very much for their positive and constructive comments and suggestions on our manuscript entitled Obtaining land cover type for urban rainwater model in UAV images using MRF and MKFCM clustering techniques. (Manuscript ID: ijgi-466696)

We have considered reviewer’s comments carefully and have made revision which marked in red in paper. We have tried our best to revise our manuscript according to the comments. We would like to express our great appreciation to you for comments on our paper. Looking forward to hearing from you.

Your sincerely,

Chengcai Zhang

Reviewer 2 Report

1.      Flying height and Ground Sampling Distance of the acquired images?

2.      How did you evaluate the accuracy of the generated DOM?

3.      Figure 2: What do you mean by internal orientation? The software estimates the Exterior Orientation Parameters (EOPs) of each image and calibrates the Interior Orientation Parameters (IOPs) of the camera.

4.      It is hard to read equations in Figure 4.

5.      Line 276: What’s the actual size of the selected area?

6.      Line 161: The three kernels has to be better explained. For example, please explain: What are xi’ and xj’? What are si and sj for k3?

7.      Line 196: What is X_Ni?

8.      Table 1: If you are using  comma two separate digits (e.g., 509,559 pixels), please use the same format in Table 1.

9.      Please show the Kappa coefficients for FCM and MKFCM-MRF.

10.  Please add line numbers for Page 14.

11.  Page 14: Kapaa -> Kappa

12.  Any explanation for why water bodies are divided into impermeable lands in MKFCM-MRF?

Author Response

(The authors gave the same response as above.)

Round 2

Reviewer 1 Report

OK

Author Response

Dear reviewer:

Thank you very much for your great efforts on our manuscript. We also appreciate your valuable suggestions and questions.

Thank you and best regards.

Yours sincerely,
Chengcai Zhang

Reviewer 2 Report

The pix4D report only reports the internal accuracy (i.e., precision) from the bundle adjustment. The accuracy I mentioned is the comparison to ground control points (GCPs) and check points. This is an external verification of the accuracy of your UAV product. Also noticed that no GCPs are used for geo-referencing. Are you only using the GPS tag from the utilized UAV for the geo-referencing? If no GCP has been used, this has to be mentioned in the manuscript. I presume that you are mainly using the derived UAV-based ortho-mosaic for classification purpose and the accuracy evaluation might not be that critical when compared to other mapping tasks. However, it is still important to report your actual configuration in the manuscript. 

Figure 4: In the previous comments, I mean that you got bad resolution when converting these equations to pictures and put them in the figure. The equations are blurred in the figure. You should have better resolution for these equation pictures.

Lines 262-263: I don't understand what this paragraph is for.

Author Response

Dear reviewer:

Thank you very much for your great efforts on our manuscript. We also appreciate your valuable suggestions and questions.

This is my first time as a corresponding author. Because I am not very familiar with the response request, so that I forgot to use the "Track Changes" function in manuscript when I uploaded the manuscript last time. I am so sorry.

This time, the manuscript with the function of "tracking changes" will be submitted.

Thank you and best regards.

Yours sincerely,
Chengcai Zhang
